# The Role of Pulmonary Surfactant Phospholipids in Fibrotic Lung Diseases

**DOI:** 10.3390/ijms24010326

**Published:** 2022-12-25

**Authors:** Beatriz Tlatelpa-Romero, Verna Cázares-Ordoñez, Luis F. Oyarzábal, Luis G. Vázquez-de-Lara

**Affiliations:** 1School of Medicine, Universidad Nacional Autónoma de México, Ciudad de Mexico 04510, Mexico; 2Complejo Regional Nororiental, Benemérita Universidad Autónoma de Puebla, Puebla 72570, Mexico; 3Laboratorio de Medicina Experimental, Facultad de Medicina, Benemérita Universidad Autónoma de Puebla, Puebla 72420, Mexico

**Keywords:** pulmonary surfactant, phospholipids, lung fibrosis

## Abstract

Diffuse parenchymal lung diseases (DPLD) or Interstitial lung diseases (ILD) are a heterogeneous group of lung conditions with common characteristics that can progress to fibrosis. Within this group of pneumonias, idiopathic pulmonary fibrosis (IPF) is considered the most common. This disease has no known cause, is devastating and has no cure. Chronic lesion of alveolar type II (ATII) cells represents a key mechanism for the development of IPF. ATII cells are specialized in the biosynthesis and secretion of pulmonary surfactant (PS), a lipid-protein complex that reduces surface tension and minimizes breathing effort. Some differences in PS composition have been reported between patients with idiopathic pulmonary disease and healthy individuals, especially regarding some specific proteins in the PS; however, few reports have been conducted on the lipid components. This review focuses on the mechanisms by which phospholipids (PLs) could be involved in the development of the fibroproliferative response.

## 1. Pulmonary Surfactant (PS)

PS is essential for life. Their main function is to lower surface tension within the alveoli and facilitate breathing. PS is a mixture of 10% specific surfactant proteins A, B, C, and D (SP-A, SP-B, SP-C and SP-D) and 90% lipids, including phospholipids (PLs), triglycerides, cholesterol, and fatty acids [1]. PLs are the major lipidic components of the PS [2,3], the most abundant being phosphatidylcholine, which represents 70–75%, half of which is found as dipalmitoyl-phosphatidylcholine (DPPC) [4,5]. This phospholipid confers the PS surface-active properties [4,6]. Phosphatidylglycerol makes up about 8% of the PLs in the PS [7]. Phosphatidylglycerol is involved in the adsorption and distribution of PS on the alveolar surface and is also a marker for lung maturity [8,9]. Phosphatidylglycerol and phosphatidylinositol (PI), the latter of which is present in 3% [7], are phospholipids that function as regulators of innate immune processes by antagonizing the cognate ligand activation of Toll-like receptor 2 (TLR2) and 4 (TLR4). Phosphatidylglycerol and phosphatidylinositol act by blocking the recognition of activating ligands by the TLRs, either directly or via the TLR4 coreceptors CD14 and MD2. This mechanism implicate reduction on protein phosphorylation events which characterize the intracellular signaling by activated TLRs and inhibition of transcription of several pro-inflammatory genes [10,11]. In vitro studies in bronchial epithelial cells showed that overexpression of TLR2 and TLR4 induced a pro-fibrotic milieu by increasing the production of the proinflammatory proteins HMGB1 or IL-27 [12,13]. In a clinical context, TLR2 and TLR4 expression were significantly increased in lungs of patients with different forms of ILD [14]. This evidence raises the importance of phosphatidylglycerol and phosphatidylinositol in the pathogenesis of TLRs in pulmonary fibrosis.

The remaining PLs consist of phosphatidylethanolamine, sphingomyelin, and phosphatidylserine [15,16,17]. Phosphatidylethanolamine is found in PS, composing 1.5% of all PLs [7,18]. Phosphatidylethanolamine has a smaller head group, which gives the lipid a cone shape, and in membranes, the acyl chains of phosphatidylethanolamine impart lateral pressure that can be released by the membrane adopting negative curvature [5,19]. Phosphatidylethanolamine exhibits antifibrotic properties by reducing collagen gene expression in human lung fibroblasts [20]. Cholesterol comprises 4% [7], which is required to maintain the structure and properties of the lipid monolayer [21]. Cardiolipin is a minor component 1.1% but may have important roles in lung homeostasis [22]. These lipid components of PS contribute to the functionality and stability of the alveolar epithelium and are mainly for the gas exchange of O_2_ and CO_2_ [6].

The molecular mechanisms related to pulmonary surfactant lipid components represent a relevant topic to understand their implications in disease. A recent study analyzed the molecular species in phosphatidylinositol, phosphatidylglycerol, and phosphatidylcholine in PS from newborn calves and cows. Results showed that in bovine PS, the profiles of molecular species for the two anionic headgroups, phosphatidylinositol and phosphatidylglycerol, are distinct. In fact, the profiles of the diacyl pairs for the two anionic headgroups could be functionally important [23]. For instance, the polyunsaturated fatty acid in phosphatidylinositol should be more susceptible to oxidation and, therefore, undesirable after the shift from placental to pulmonary gas exchange. In contrast to phosphatidylinositol, phosphatidylglycerol and phosphatidylcholine contain similar diacyl pairs. The common set of molecular species in phosphatidylglycerol and phosphatidylcholine, however, may represent a set of compounds in the mature lung. Although those results agree with determinations of the anionic molecular species in PS from several animals, in human PS, for instance, the diacyl pairs correlate well between the two anionic headgroups, both in terms of the compounds and their relative amounts, suggesting that the difference between the two sets of anionic molecular species is common but not universal. It is probable that shifts of the anionic headgroup may also reflect changes in the diacyl pairs in other circumstances, as in pulmonary disease. In both the acute injury of acute respiratory distress syndrome (ARDS) and the more chronic injuries of pulmonary fibrotic diseases, the ratio of phosphatidylinositol to phosphatidylglycerol is consistently increased relative to PS from normal lungs. This change may be associated with the reversion of the ATII cells to a more juvenile phenotype, but the shift may produce functional changes [23].

### 1.1. Pulmonary Surfactant Lipids

#### 1.1.1. Lipid Biosynthesis Pathways

PS and cell membrane PLs are synthesized through the same biosynthetic pathways [4]. The most abundant PLs in pulmonary surfactants are those composed of a glycerol skeleton with a polar phosphate group and two esterified fatty acids at *sn*-1 and *sn*-2 positions [2]. PS biosynthesis is critically dependent on the availability of fatty acids. Fatty acid sources are triglycerides carried in lipoproteins [24], PS reuptake and fatty acid de novo biosynthesis [7]. The keratinocyte growth factor (FGF) stimulates key enzymes involved in fatty acid biosyntheses, such as acetyl CoA carboxylase, fatty acid synthase and citrate lyase that are important for maintaining ATII cells and restoring the epithelium after lung injury [25,26].

Phosphatidic acid (PA) is an intermediate of phospholipid biosynthesis and a key precursor in PS biosynthesis [27]. PA de novo synthesis utilizes dihydroxyacetone phosphate (DHAP), which can be processed by two different pathways: the glycerol-3-phosphate (G3P) pathway that occurs in mitochondria and endoplasmic reticulum and the dihydroxyacetone phosphate (DHAP) pathway that occurs in endoplasmic reticulum and peroxisomes [28]. In the G3P pathway, G3P is generated directly from DHAP by the enzyme glycerol-3-phosphate dehydrogenase (G3PD), a reaction that requires nicotinamide adenine dinucleotide (NADH), or conversion of glycerol directly to G3P [2]. In the synthesis of phosphatidylcholine, phosphatidylglycerol and phosphatidylinositol, precursors of DHAP, G3P, PA, and choline are used as well as some acyl derivatives made from these precursors [5,8,29]. The enzymes for phosphatidylcholine biosynthesis are choline kinase, CTP: phosphocholine cytidylyltransferase (CCT), choline phosphotransferase (CPT), acyltransferase and enzymes of the fatty acid biosynthesis [30,31]. The genes Pcyt1a and Pcyt1b encoded multiple CCT isoforms, CCTα, CCTb2, and CCTb3, all of them expressed ubiquitously. CCTα is the isoform most expressed in ATII cells [32,33]. CCT is the main regulatory enzyme for the de novo phosphatidylcholine synthesis in lung and ATII cells [34]. DPPC, a major lipid in the PS, is synthesized by the enzyme carnitine palmitoyltransferase and disaturated diacylglycerol [35]. Saturated phosphatidylcholine is synthesized by the de novo Kennedy pathway or by the Lands cycle from unsaturated phosphatidylcholine [36]. Unsaturated phosphatidylcholine is deacetylated by the enzyme phospholipase 2 (PL2) to produce lysophosphatidylcholine, then the enzyme lysophosphatidylcholine acyltransferase 1 reacylates lysophosphatidylcholine with a saturated fatty acid to produce phosphatidylcholine [37]. The newly synthesized phosphatidylcholine moves from the smooth endoplasmic reticulum to the lamellar bodies (LBs) for storage prior to PS secretion [38].

Phosphatidylglycerol is synthesized from CDP-diacylglycerol, from PA and CTP by an endoplasmic reticulum membrane-associated CDP-diacylglycerol synthase [39]. CDP-diacylglycerol by the enzyme glycerophosphate phosphatidyltransferase is converted to phosphatidylglycerolphosphate, followed by its dephosphorylation to phosphatidylglycerol by the enzyme phosphatidylglycerophosphatase [40].

Phosphatidylethanolamine synthesis involves the conversion of ethanolamine to phosphoethanolamine, a chemical reaction catalyzed by the enzyme ethanolamine kinase. Then, phosphoethanolamine is converted to cytidine-diphosphoethanolamine by the enzyme CTP: phosphoethanolamine cytidylyltransferase [41]. The final step of phosphoethanolamine formation is catalyzed by ethanolamine phosphotransferase [42].

Phosphatidylinositol is synthesized from CDP-diacylglycerol [37], then CDP-diacylglycerol: inositol-3-phosphatidyltransferase catalyzed the direct reaction of CDP-diacylglycerol to myo-inositol or phosphatidylinositol synthase that is also localized in the endoplasmatic reticulum of ATII cells [43].

Phosphatidylserine is synthesized from a base exchange reaction of phosphatidylcholine or phosphoethanolamine via phosphatidylserine synthases(PSS1 and PSS2), which is carried out in the endoplasmic reticulum and/or mitochondrial membranes in ATII cells [42].

Cardiolipin is synthesized by the condensation of phosphatidylglycerol and CDP-diacylglycerol, a reaction catalyzed by cardiolipin synthase [44]. Cardiolipin contains 18:2 acyl chains from phosphatidylcholine and/or phosphoethanolamine donors in a transacylation or remodeling reaction mediated by the protein tafazzin [45].

Sphingomyelin synthesis involves the condensation of L-serine with palmitoyl-CoA to produce 3-ketosphinganin, a reaction catalyzed by serine palmitoyltransferase. The 3-ketosphinganine product is reduced to sphinganine [46]. The final step in this pathway is the transformation of a phosphatidylcholine head group to ceramide-producing Sphingomyelin, which is catalyzed by sphingomyelin synthase [47].

#### 1.1.2. Pulmonary Surfactant Metabolism: The Role of Lipids

Following the synthesis of lipids and PS proteins in the endoplasmatic reticulum, PS components are assembled in intracellular storage organelles, called lamellar bodies (LB) and are subsequently secreted into the alveolar space to decrease surface tension [48]. For this to occur, phosphatidylcholine, phosphatidylglycerol and cholesterol must move into the lumen through the membrane of the limiting LB, generating the tightly packed membranous structure [49,50]. This is done with the help of a transporter called ATP-binding cassette-3 (ABCA3), which is expressed in the membrane of LBs [51,52].

LBs are secreted into the aqueous phase, where other extracellular forms of PS appear in the form of tubular myelin that can be efficiently adsorbed at the air-water interface to form a stable PS film [15,48]. Then a large part of the PS is absorbed by alveolar epithelial ATII cells, and the components that were not recycled will be degraded by macrophages [48]. Granulocyte-macrophage colony-stimulating factor (GM-CSF) is necessary for the ability of macrophages to remove PS that is to be discarded [53,54].

#### 1.1.3. The Role of Phospholipids in Lung Surfactant Biophysics

The biophysical properties and extracellular metabolism of the PS are complex, and the exact mechanisms by which PS reduces surface tension during alveolar expansion and withstands high levels of pressure during lateral compression at the end of expiration remains unclear [6]. On expiration, PS organizes into multilayer structures characterized by the presence of DPPC at the air-liquid interface with a consequent drastic reduction of surface tension (<2 mN/m); during inspiration: re-extension of phospholipid membranes together with initial adsorption of PS aggregates at the air-liquid interface, reaching the equilibrium surface tension of 20 mN/m. This process is mediated by SP-A and SP-B, in addition to being facilitated by phosphatidylglycerol, cholesterol and unsaturated phospholipids [1,6]. In the alveolar lumen, PS reduces the surface tension on the alveolar walls from 70 mN/m to almost 1 mN/m [1].

## 2. Interstitial Lung Diseases

Interstitial lung disease is the name for a large group of lung disorders that cause fibrosis of the lungs. Over time, fibrosis can cause lung stiffness and eventually affect breathing. There are many different substances, conditions and triggers that can lead to interstitial lung disease. Below we briefly present some pulmonary diseases in which surfactant components are altered.

### 2.1. ILD of Know Cause: Occupational Exposure

#### 2.1.1. Asbestosis

Asbestosis is pneumoconiosis caused by inhaling asbestos fibers, which elicit potent inflammatory responses in the respiratory tract [55]. It has been shown that asbestos can cause lipid peroxidation and trigger genomic alterations in pulmonary cells [56]. Furthermore, a decrease in phospholipase C (PLC) activity has been described, which directly impacts the biosynthesis of lipids like phosphatidylinositol, phosphatidylglycerol and phosphatidylserine, affecting macrophage activity and PS availability [57]. In asbestosis, alterations in the alveolar activity of CC-16 (specific proteins of clear cells), SP-A, and phospholipase A2 (PLA2) have been reported. [58]. The activity of PLA2 secreted in alveolar fluids and the effect of CC-16 on platelet growth factor-induced chemotaxis of human fibroblast [59]. SP-A is associated with lung diseases such as IPF and alveolar proteinosis [60], while increased PLA2 has been implicated in inflammatory lung disease pathologies such as respiratory distress syndrome, asthma and alveolar proteinosis [59]. A consequence of asbestos parenchymal damage is pulmonary fibrosis. The molecular mechanisms by which it could occur are primarily by the death of ATII cells mediated by mitochondrial and p53 death pathways resulting from the production of iron-derived reactive oxygen species (ROS) and DNA damage [61]. In the development of pulmonary fibrosis caused by asbestos, the death of ATII and the decrease in macrophage activity could be involved (Figure 1).

#### 2.1.2. Silicosis

Occupational exposure to crystalline-free silica SiO_2_ dust causes silicon nodules and diffuse pulmonary fibrosis. Lung function is severely damaged by silicosis fibrosis, whose degree increases over time [62]. The alveoli of lungs affected by acute silicosis are lined by hypertrophic ATII, which produce proteinaceous materials, and PS is produced, resulting in protein-filled alveoli [63,64]. A high level of PS components in the lungs is a result of exposure to silica dust [65]. Disaturated phosphatidylcholine and SP-A levels increased in intracellular and extracellular PS compartments following silica exposure in rats [66]. A high concentration of silica reduces the number of alveolar macrophages, which are responsible for decreased DPPC liposomes during PS metabolism [67] (Figure 2).

### 2.2. ILD of Know Cause: Related to Treatment

#### 2.2.1. Radiation

Radiation therapy (also called radiotherapy) kills cancer cells and shrinks tumors by administering high doses of radiation. However, the use of radiotherapy can cause acute or chronic pneumonia [68]. The inflammatory process of acute pneumonia involves the formation of intra-alveolar and septal edema, as well as the formation of epithelial and endothelial desquamations. During acute pneumonia, ATII cells play a critical role in increasing microvascular permeability. This hypothesis is supported by the findings that ATII is the primary cellular response to lung injury. Then, a secondary response of ATII cell hyperplasia occurs as a result of the plasma protein overload on lymphatic and other drainage mechanisms. Consequently, an excess of PS results, causing a fall in compliance, abnormal gas exchange values, and even respiratory failure. As a result of the inflammatory process, chronic fibrosis can develop [69]. Other studies reveal that radiation causes apoptosis of epithelial cells, as well as a large amount of oxidative damage and release of fibrotic cytokines [70]. Oxidative damage affects lipid metabolism, increases inflammatory cytokines such as interferon-γ (INF-γ) and tumor necrosis factor-α (TNF-α) by macrophages [71]. Radiation increases glucose catabolism by increasing the levels of glycerophosphate as a lipid precursor [72], increases the expression of lipoproteins lipase and fatty acid binding protein (FABP4) and triacylglycerides, all resulting in lipid accumulation [73] (Figure 3).

#### 2.2.2. Amiodarone

Amiodarone is a highly toxic antiarrhythmic drug that contains iodinated benzofuran, which causes fibrosis in the lungs [74]. Several studies have demonstrated that the accumulation of multilamellar bodies in the cytoplasm is the result of reduced degradation of PLs since amiodarone inhibits lysosomal phospholipases A1 and A2 [75,76]. Furthermore, amiodarone and the metabolite desethylamiodarone had a high capacity to bind to DPPC and prevent the reuptake of LBs [76]. Therefore, phospholipid accumulation and oxidative stress may contribute to pulmonary fibrosis through amiodarone exposure, so new therapies are aimed at modulating lipid homeostasis (Figure 4).

## 3. Granulomatous Diseases

### 3.1. Sarcoidosis

Sarcoidosis is a pulmonary disease characterized by granulomas formed by mononuclear phagocytic cells in the alveolar, bronchial and vascular walls [77]. In pulmonary lavages of people diagnosed with sarcoidosis, it has been found a decrease in phospholipids such as phosphatidylcholine, which is the main component of phosphatidylserine [78]. High expression of peroxiredoxin has been found in granulomas [79] which expresses phospholipase A2 activity that is Ca^2+^ independent. Phospholipase A2 activity is important for the degradation of internalized phosphatidylcholine and its resynthesis by the reacylation pathway [80]. Increased phospholipase A2 levels are associated with PS diffusion resulting in inflammatory lesions in the lung [81].

### 3.2. Hypersensitivity Pneumonitis

Hypersensitivity pneumonitis is a disease caused by the reaction of the immune system to airborne and inhaled substances [82]. There are two types of hypersensitivity pneumonitis, non-fibrotic pneumonitis, associated with chronic bronchiolocentric interstitial inflammatory infiltrate and sometimes associated with granulomas or giant cells, and fibrotic pneumonitis takes the form of interstitial fibrosis confined to peribronchiolar regions or non-specific interstitial pneumonia and with more granulomas than non-fibrotic pneumonitis [83]. In hypersensitivity pneumonitis, increased sphingomyelin and phospholipid imbalance have been documented. This change suppresses the activity of PS in the activation and proliferation of lymphocytes in the lung [78]. Sphingomyelin metabolism generates ceramide, which inhibits phosphatidylcholine synthesis [84], is involved in the inflammatory response and is accumulated in alveolar cells [85], which explains the decrease of the rest of the PS components in hypersensitivity pneumonitis.

## 4. ILD of Unknow Cause: Idiopathic Interstitial Pneumonias

### Idiopathic Pulmonary Fibrosis

IPF is a chronic, progressive disease of unknown origin, without effective treatment and highly fatal [86,87]. IPF is described by the destruction of the alveolar epithelium and reduction of the airspace occupied by fibroblasts that form fibrotic foci characterized by having an accumulation of collagen-rich extracellular matrix [88]. Dysfunction and apoptosis of ATII cells initiate the onset and progression of IPF [89]. Apoptosis and depletion of ATII cells in IPF leads to decreased synthesis of PS, which is indispensable to prevent the growth of other cells, such as fibroblasts that are recruited after epithelial damage [90]. Apoptosis is associated with changes in phospholipid metabolism, releasing fatty acids, such as arachidonic acid, which is paralleled by a reduction in cell viability [91]. This could occur in ATII cells [92]. After repetitive damage to membranes, there is the activation of enzymes such as phospholipases that hydrolyze ester bonds present in phospholipids [24]. Products of this are lysophospholipids, such as lysophosphatidic acid increased in IPF patients [93]. Lysophosphatidic acid mediates the fibrotic scarring process [94], prevents proper epithelial regeneration, increases fibroblast recruitment and promotes fibroblast resistance to apoptosis [95]. The overgrowth of fibroblasts leads to collagen accumulation and activation of growth factors such as transforming growth factor β (TGF-β) [96]. TGF-β is a potent profibrotic cytokine that mediates its fibroproliferative effects by inducing apoptosis of ATII cells [97]. Laboratory studies show that PS induces a Ca^2+^ signal associated with apoptosis in fibroblasts of normal human lungs. Activation of G protein-coupled receptors (GPCRs) by an extracellular ligand activates phospholipase Cβ (PLCβ), resulting in subsequent cleavage of the membrane phospholipid phosphatidylinositol 4,5-bisphosphate into diacylglycerol and IP_3_, the latter penetrating the endoplasmic reticulum and causes the release of intracellular Ca^2+^ [20,98]. This proposes a targeted therapy (Figure 5).

## 5. Pulmonary Fibrosis Secondary to COVID-19 and ARDS

Acute Respiratory Distress Syndrome (ARDS) is mainly caused by pneumonias of bacterial and viral origin, as well as less frequently by severe sepsis and extrapulmonary trauma [99]. Due to the SARS-CoV-2 pandemic, the incidence of ARDS has increased as many of these patients progressed to severe forms of COVID-19 with respiratory dysfunction [100]. In ARDS, an imbalance between profibrotic and antifibrotic factors occurs due to the accumulation of macrophages, fibrocytes, fibroblasts and myofibroblasts in the alveoli, which favor the production of fibronectin, collagen and other components of the extracellular matrix, driving a mechanism of pathological fibroproliferation that results in lungs with fibrotic changes of varying degree, even in early stages of the disease [101]. Pulmonary fibrosis is a well-recognized complication of ARDS, and there is evidence that patients who survive severe forms of COVID-19 have signs of pulmonary fibrosis that persist over time and is related to the presence of ARDS during its evolution [101,102,103].

A recent study in patients diagnosed with ARDS showed that serum autotaxin (ATX) levels correlated with the severity of lung injury, while the concentration in bronchoalveolar lavage fluid was positively associated with fibrotic mediators such as matrix metalloproteinase-7 (MMP-7), fibronectin, oncostatin M (OSM) and secreted protein acidic and in cysteine (SPARC), suggesting that ATX may be related to the development of pulmonary fibrosis in patients with ARDS [104]. Currently, in the Laboratory of Experimental Medicine of the *Benemérita Universidad Autónoma de Puebla,* we are testing the hypothesis that lysophosphatidic acid is associated with the development of pulmonary fibrosis in patients with ARDS.

## 6. Conclusions

The fundamental role of lung surfactant PLs is to participate in the biophysical properties of PS; however, other functions have recently been described. This review summarizes evidence on the potential role of LPs of PS in the physiopathology of fibrosing lung diseases.

The ATX, phosphatidylethanolamine, lysophosphatidic acid and their receptor pathways offer a new approach to understanding the pathogenesis of idiopathic pulmonary fibrosis and pulmonary fibrosis because of ARDS. Moreover, they are currently targets for the development of new drugs against these diseases.

## Figures and Tables

**Figure 1 ijms-24-00326-f001:**
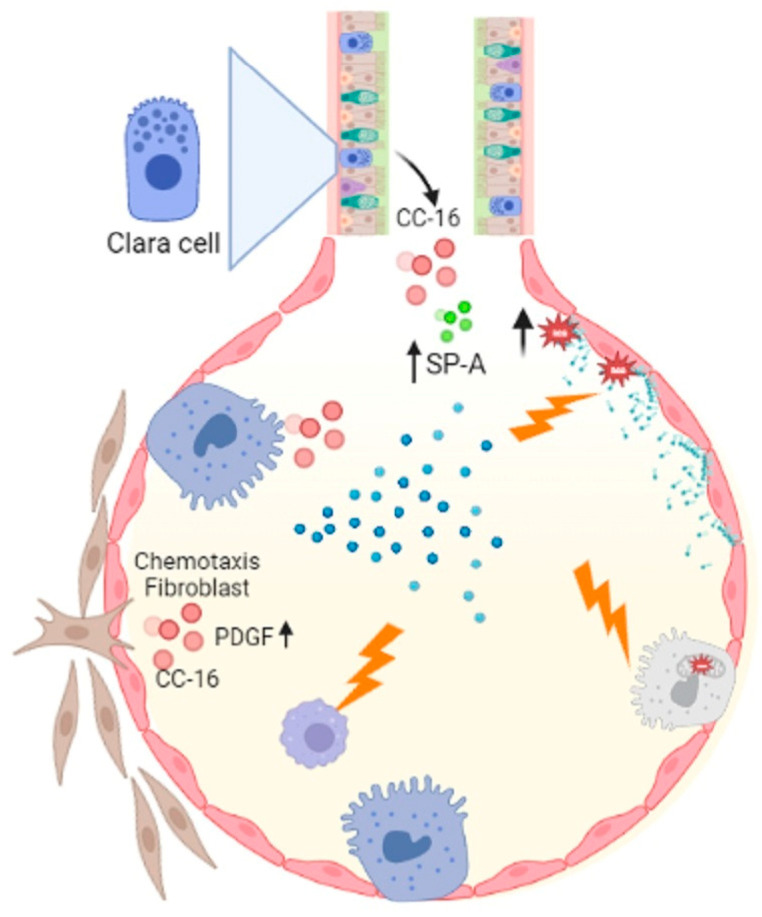
Asbestosis influences macrophage activity which is important for eliminating cellular debris, and acts on ATII cells by activating the mitochondrial and p53 death pathways. Asbestos catalyzes lipid peroxidation by decreasing PLC, which decreases phosphatidylinositol, phosphatidylglycerol, and phosphatidylserine synthesis. Another important event is the alteration of CC-16 with fibroblast chemotaxis activity, increased Sp-A and PLA2. Figure created in BioRender.com.

**Figure 2 ijms-24-00326-f002:**
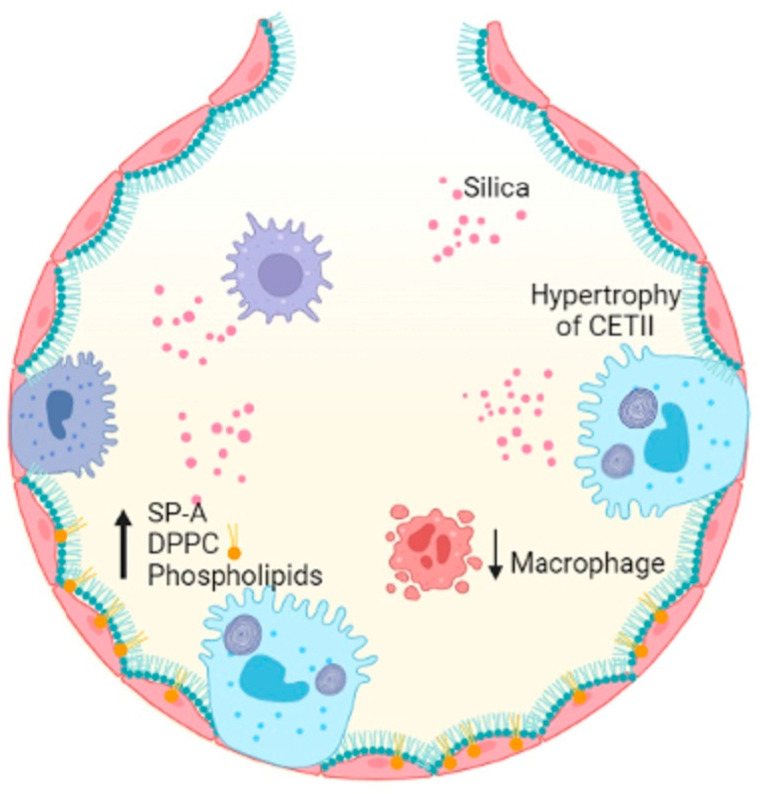
Silica entering the alveolar space decreases the number of macrophages, induces hypertrophy of ATII cells and results in increased SP-A, DPPC and PLs, damaging alveolar homeostasis. Figure created in BioRender.com.

**Figure 3 ijms-24-00326-f003:**
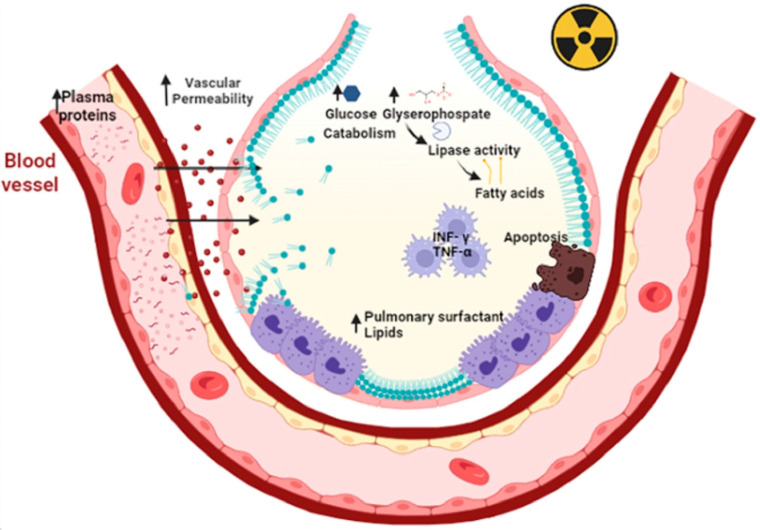
Primary radiation increases microvascular permeability and plasma proteins that overwhelm the drainage mechanisms. This elicits a secondary response to hyperplasia of type II epithelial cells. In addition, there is an accumulation of pulmonary surfactant within the alveolus that triggers apoptosis of type II epithelial cells, oxidative damage and the release of fibrotic cytokines such as TNF-Υ and TNF-α. Primary radiation increases the catabolism of lipoprotein lipases due to increased glucose and glycerophosphate. This results in the release of fatty acids leading to decreased distensibility and respiratory failure. Figure created in BioRender.com.

**Figure 4 ijms-24-00326-f004:**
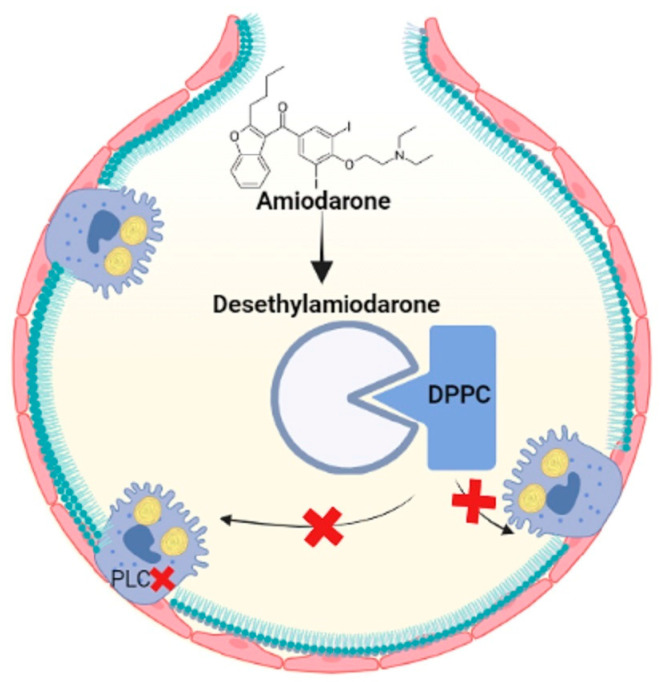
Amiodarone is metabolized to desethylamiodarone, which binds to DPPC, prevents it from being taken up by ATII cells and inhibits phospholipases and increases lipids in the alveolar space. Figure created in BioRender.com.

**Figure 5 ijms-24-00326-f005:**
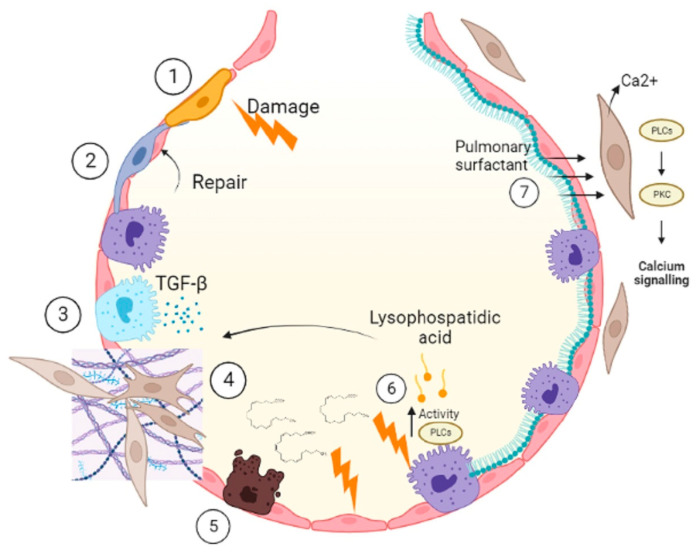
During IPF, damage occurs in the epithelium (**1**); normally, when there is a lesion, the ATII cells recover the epithelium (**2**). In IPF, the ATII cells respond in an aberrant manner which means that they secrete cytokines (**3**) that provoke migration, proliferation and differentiation of fibroblasts (**4**). These fibroblasts secrete extracellular matrix and change the architecture of the alveolus, causing it to lose its function. The damage to the ATII cells is progressive, and there is a change in the metabolism of PLs. During this process, fatty acids such as arachidonic acid are released, which decreases the viability of epithelial cells (**5**), and also increases the activity of phospholipases which produces palmitic acid that mediates scarring in the fibrotic process, prevents the regeneration of the epithelium and increases the recruitment of fibroblasts resistant to apoptosis (**6**). PS is important for maintaining alveolar homeostasis and inducing apoptosis of cells such as fibroblasts in which PS induces a calcium signal (**7**). Figure created in BioRender.com.

## Data Availability

Not applicable.

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
