# Peer review of "The Role of Pulmonary Surfactant Phospholipids in Fibrotic Lung Diseases"

_ijms, 2022, doi:10.3390/ijms24010326_

Round 1

Reviewer 1 Report

In this review Tlatelpa-Romero et al. discuss evidence in the literature about the possible involvement of the phospholipids in pulmonary surfactant in fibrotic lung diseases. The manuscript requires extensive editing, and the authors are strongly urged to seek the assistance of a native English speaker to improve the English language usage. In addition, there are some early inaccuracies and disparities that might cause readers to call into question the utility of the review if the errors are not corrected. Some examples include:

(1)   Dipalmitoylphosphatidylcholine (DP-PC) is simply one particular species of phosphatidylcholine (PC, which comprises multiple species), not two different lipids.

(2)   In line 31, how can phosphatidylglycerol (PG) make up about 10% of the surfactant phospholipids but then in lines 36-37 PG plus phosphatidylinositol (PI) is 2-10% of total phospholipids? Also are these percentages in humans or in animal models?

(3)   In lines 278-279, the authors have the incorrect order: first the GPCR is activated and then activates PLCbeta but perhaps what they were trying to say is: “…this effect is mediated by a phospholipase Cbeta (PLCbeta)-induced Ca2+ signal following activation of G-protein-coupled receptors,…..”

Other issues that need to be addressed include:

(4)   All abbreviations should be defined on first use.

(5)   Lines 97-101 describing the generation of CDP-DAG should occur in the paragraph above (lines 86-89) where CDP-DAG is first mentioned. Also in line 100 what is PI synthase and does its mention belong here in the sentence?

(6)   Lines 151-152 do not form a sentence. Also in the paragraph in lines 151-159 and that in lines 160-167, there is redundant information. The authors should edit these two paragraphs to remove the redundancy.

(7)   In line 293-295, it is not clear why it is important to induce apoptosis. The authors should expand slightly on this idea.

Finally, the information about the ability of PG and PI to inhibit toll-like receptor activation seems quite important in terms of limiting inflammation, which may play a role in lung fibrosis. The authors are encouraged to expand on this information.

Reviewer 2 Report

The manuscript "The role of pulmonary surfactant phospholipids in fibrotic lung diseases" by Tlatelpa-Romero et al. attempts to review the literature on this relatively specific topic. It is beyond doubt that fibrotic lung diseases constitute a very important topic, and the purpose of the authors, to focus " on the mechanisms by which phospholipids could be involved in the development of the fibroproliferative response", is a worthy one.

However - and this is my main criticism of this paper -, the manuscript reads like a sequential enumeration and general description of different diseases under this umbrella, followed by a mention of studies reporting phospholipid alterations associated to those pathologies. The mechanistic part is underdeveloped. Even though some schemes are proposed (and translated into cartoons), they still look as rather phenomenon-based and the rationale between the observed relationships remains undescribed. I wish the authors would be more critical and invest more in the actual rationalization of these observations - why these particular lipid species could be involved in these diseases - in relation to their structures/functions as pulmonary surfactant components.

Additionally, the are serious deficiencies in the scholarly presentation of the manuscript, namely:

- Substandard level of written English, which in some cases makes the sentence impossible to understand (e.g., "The development of fibrosis in asbestosis could be mediated by CC-16 which may contribute to platelet-derived growth factor-induced chemotaxis and fibroblast recruitment, in addition, alteration of PS-A and PLA has also been seen another important phospholipase in the biosynthesis of PI, PS, PG is PLC which has decreased activity"; lack of punctuation in "non-fibrotic pneumonitis associated with chronic bronchiolocentric interstitial inflammatory infiltrate and sometimes associated with granulomas or giant cells and fibrotic pneumonitis takes the form of interstitial fibrosis confined to peribronchiolar regions or non-specific interstitial pneumonia and with more granulomas than non-fibrotic pneumonitis").

- Low figure resolution, which becomes clear when increasing the Zoom trying to read the labels - they are effectively unreadable;

- Confusing organization of headings - there is no "3." section, it begins straight with "3.1 Granulomatous diseases", and the same holds for the "4." section (it begins with "4.1 ILD of unknow cause: Idiopathic interstitial pneumonias"). For the reader, this seriously undermines the perceived organization of the paper.

- Lack of definition of some abbreviations (examples too numerous to enumerate here), use of the abbreviation prior to the definition, or incorrect abbreviation (e.g. line 297, "Acute Respiratory Failure Syndrome (ARDS)" - why not ARFS, then? The fact is that it is known as "Acute Respiratory Distress Syndrome", hence ARDS; LC or CL, presumably for lamellar bodies; PS or SP).

- Incorrect writing of chemical formulae, with atom indices not written in subscript font (e.g. "CO2") or charges not written in superscript ("Ca2+").

Other comments:

Lines 53-54: "Phospholipids consists [sic] of glycerol, two fatty acid tails and a head

with a phosphate group attached to an alcohol" - the group attached to the phosphate (and I don't mean the glycerol, which is previously mentioned) is not necessarily an alcohol. This may be the case for PE, but not for other phospholipids.

Line 180: "Desaturated phosphatidylcholine (DPPC)" - what is meant by "desaturated"? DPPC has two saturated acyl chains. "Disaturated" perhaps?

Lines 218-219: "Iodinated benzofuran, the component of amiodarone, is a highly toxic antiarrhythmic drug that causes fibrosis in the lungs" - a conceptual clarification is due here: strictly, the drug is amiodarone, not the iodinated benzofuran moiety.

Line 280: " inositol polyphosphate 5-phosphatase (PIP2)" - this abbreviation may be confused with that of the phospholipid, phosphatidylinositol 4,5-bisphosphate.

Round 2

Reviewer 2 Report

The authors have revised extensively and thus much improved the manuscript. My sole reservation lies in the resolution of the figures, which appears to be insufficient in my view.